# Visual Jenga: Discovering Object Dependencies via Counterfactual Inpainting

**Anand Bhattad**[1]*    **Konpat Preechakul**[2]    **Alexei A. Efros**[2]

[1]Johns Hopkins University    [2]University of California, Berkeley

https://visualjenga.github.io

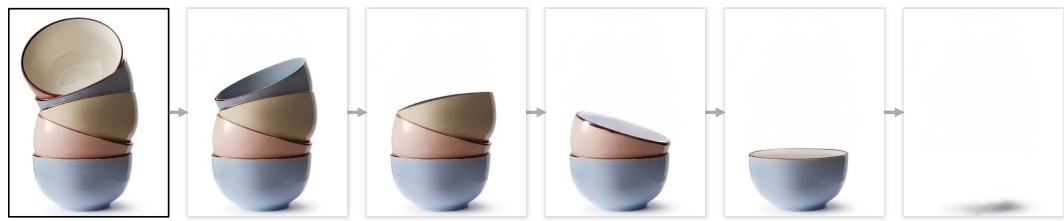

Figure 1: **Visual Jenga:** Given an input image (left), we generate a sequence of images, removing one object at a time while keeping the scene stable. We argue that this new task provides a useful signal to assess the level of grounded scene understanding of vision systems. *See the video on our project page for animated results.*

## Abstract

This paper proposes a novel scene understanding task called Visual Jenga. Drawing inspiration from the game Jenga, the proposed task involves progressively removing objects from a single image until only the background remains. Just as Jenga players must understand structural dependencies to maintain tower stability, our task reveals the intrinsic relationships between scene elements by systematically exploring which objects can be removed while preserving scene coherence in both physical and geometric sense. As a starting point for tackling the Visual Jenga task, we propose a simple, data-driven, training-free approach that is surprisingly effective on a range of real-world images. The principle behind our approach is to utilize the asymmetry in the pairwise relationships between objects within a scene and employ a large inpainting model to generate a set of counterfactuals to quantify the asymmetry.

## 1   Introduction

Can one truly understand a scene by simply naming the objects in it? While modern computer vision methods excel at object detection and semantic segmentation, these capabilities often prove inadequate for practical purposes, such as vision-guided robot manipulation or truly grounded image editing. Treating scenes as static collections of isolated elements, recognition models neglect the critical relationships between objects that give scenes their intrinsic meaning. In this paper, we argue that true *scene understanding* necessitates understanding how objects depend on and interact with one another within the space of a scene.

Drawing inspiration from the game Jenga[1], we propose a novel task: to virtually deconstruct scenes by carefully erasing objects, much like players strategically remove blocks from a tower while making

---

*work done while AB was a RAP at Toyota Technological Institute at Chicago

[1]Jenga is derived from the Swahili word "kujenga" meaning "to build". There are similar games in other cultures: pick-up sticks, mikado, jonchets, etc.

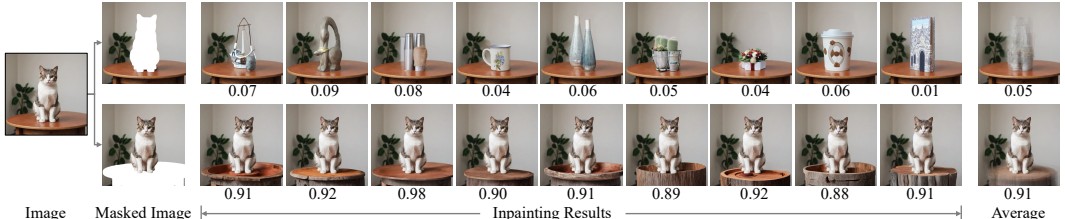

| | | 0.07 | 0.09 | 0.08 | 0.04 | 0.06 | 0.05 | 0.04 | 0.06 | 0.01 | 0.05 |
| | | 0.91 | 0.92 | 0.98 | 0.90 | 0.91 | 0.89 | 0.92 | 0.88 | 0.91 | 0.91 |

Image    Masked Image ├─────────────────── Inpainting Results ───────────────────┤ Average

Figure 2: **Counterfactual Inpainting.** For a pair of objects in an image (e.g., a cat and a table), we determine which object depends on the other by removing each object (masked images) and using an inpainting model to generate $N$ inpaintings for the masked areas. The number below each inpainting result is the cosine similarity (from CLIP and DINO) between it and the original image. The average images (for illustration only) show that the cat can be replaced by various objects, while the table remains largely stable, indicating that the table supports the cat.

sure it does not collapse. As shown in Fig. 1, the goal of Visual Jenga is to progressively remove objects from a single image, one at a time, such that the scene always remains "well-formed". Solving this task reveals the relationships between scene elements by systematically exploring which objects can be removed while preserving scene coherence both physically and geometrically.

By framing scene understanding as a sequential deconstruction task, Visual Jenga allows us to evaluate how objects relate to and depend on one another: an aspect central to scene understanding in humans [9] yet largely overlooked by current benchmarks. Such understanding is crucial in many practical domains. For instance, the ability to remove objects without destabilizing a scene is essential for many robotic manipulation tasks [42, 18]. Preserving physical scene coherence is also important for realistic image editing applications.

As a starting point for solving the Visual Jenga task, we propose a simple, data-driven approach that is surprisingly effective in a range of real-world scenes without requiring any explicit physical reasoning. Our approach uses a form of counterfactual reasoning by asking "what if this object were removed?" The principle behind our approach is using the *asymmetry* in the pairwise relationships between objects within a scene [46].

Consider the cat sitting on a table in Fig. 2. The cat depends on the table below for structural support, but not vice versa. If we consider the counterfactual of removing the table from the image, the cat would require some other support surface for the scene to remain stable [9]. If, on the other hand, we remove the cat, the scene is already stable, so it doesn't much matter what, if anything, will go in the cat's place. To make this intuition quantifiable, we use an off-the-shelf large image inpainting model to help us estimate the conditional probabilities of counterfactual images. This does not require any training and exploits existing knowledge of what constitutes a well-formed scene [9] already captured in large generative models. As Fig. 2 shows, inpainting the region occupied by the cat results in a diverse range of plausible objects that could replace it, whereas inpainting the table consistently produces similar support structures for the cat. Averaging these differences over multiple inpainting passes allows us to quantify this asymmetry and determine which object should be removed first.

In summary, our contributions are: (1) Visual Jenga task: a novel scene understanding task that evaluates object dependencies through sequential removal, inspired by counterfactual reasoning. (2) Counterfactual Inpainting approach: a training-free method that quantifies object dependencies, exploiting asymmetry in object co-occurrences using large-scale generative inpainting models. We demonstrate our approach through a quantitative pairwise evaluation as well as qualitative, full-scene decompositions.

## 2 Related work

At the dawn of computer vision in the 1960s, when perception and action were considered two sides of the same coin, the grand goal of image understanding was the ability to reason about the physical scene from an image. Roberts' BlocksWorld [63], the very first PhD thesis in computer vision, was all about analyzing object relationships within a physical scene (made up of simple blocks) so that a robot could pick up these blocks one-by-one and reassemble them into a different configuration. Alas, in the 60 years that followed, the goal of image understanding has been watered down to a

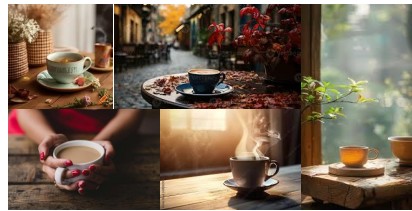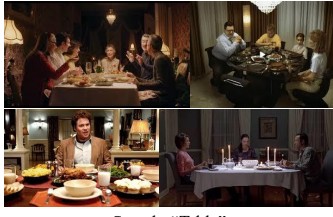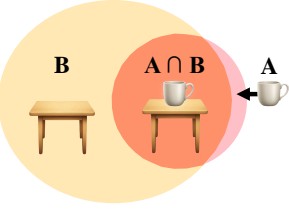

| Search: "Cup" | Search: "Table" | $P(\text{Table}|\text{Cup}) \gg P(\text{Cup}|\text{Table})$ |

Figure 3: **Asymmetric Relationships in Real-World Images.** Consider performing two internet image searches: "cup" (left) and "table" (middle). Notice that almost all the cups are depicted on top of a table, whereas images of tables rarely contain cups. The Venn diagram (right) illustrates this relationship: observing a table ($B$) does not guarantee a cup ($A$), but observing a cup ($A$) strongly implies a table ($B$). That is, $P(\text{Table} \mid \text{Cup}) \gg P(\text{Cup} \mid \text{Table})$. We leverage these asymmetric relationships to infer object dependencies in a scene from the distributions $P(A \mid B)$ and $P(B \mid A)$ learned from large-scale data.

combination of object detection and semantic "segmentation" [37], and nowadays, image captioning. We now review prior work that considered the task of scene understanding in its original meaning.

**Qualitative 3D scene understanding.** Psychologist Irving Biederman's classic work on scene perception [9] argues that the way humans interpret visual scenes goes far beyond a list of objects or a text description. Biederman identified several physical and geometric relational constraints between scene objects (such as physical support and occlusion) that must be satisfied for a scene to be well-formed. Inspired by this, Hoiem et al. focused on incorporating Biederman's constraints into their scene understanding systems [31, 32, 66, 26]. Subsequent research built on this by developing layered scene representations via layer-wise decomposition [34, 79], object-level deocclusion [45], and multi-layer reasoning [16]. More recently, physics-aware scene understanding inspired by the original BlocksWorld [63] has been revisited, both in synthetic settings [40, 42], as well as in attempts to generalize it to real-world scenes [27, 48, 67, 69, 68].

**Counterfactual spatial reasoning.** Identifying causal relationships in the real world from observation has been an open problem in the causal inference community [49]. In computer vision, Lopez-Paz et al. [46] have considered the special case of an object causing the presence of another object using *causal disposition* as a measure. Goyal et al. [24] use visual interventions to explain model decisions by showing how modifying specific image elements alters predictions. Besserve et al. [6] introduced counterfactual interventions to pinpoint modular components in generative networks, enabling targeted image editing and causal analysis of internal representations. Zhou et al. [80] evaluates human responses to synthetic block tower simulations to understand how people assess physical support relations. Our work scales up the principles laid by these works to complex real-world data using large pretrained generative models. While observational data may only provide statistical co-occurrence information, which is not truly causal, large-scale models trained on images [28, 3] and videos [5] show impressive counterfactual modeling capabilities, via text prompting [10], visual prompting [4], and even simple classification [41]. The underlying visual understanding of a well-formed scene in generative models covers a wide range of attributes, like geometry, materials, lighting and support, among others [78, 7, 17, 8, 73], and has shown promise for identifying object segmentation [51] and even amodal segmentation [54].

**Object dependencies and scene graphs.** Another attempt at deeper scene understanding is a line of work representing a scene as a graph of relations between atomic units, such as objects. Visual Memex [47], an early work in this area, treated each object as a node with edges for visual similarity, spatial co-occurrences, etc. Visual Genome [38] extended this to object categories using a large-scale crowd-sourced scene graph, spurring further research [77, 74]. However, these methods focus on 2D relationships and neglect geometric or physical aspects. Other efforts factorize images into object-centric latent components [12, 25, 56, 19, 20], but they treat objects independently, missing interactions and support relations.

**Object removal.** Prior work has largely focused on evaluating the visual quality of object removal, insertion, and inpainting [71, 13, 81]. Existing benchmarks assess inpainting quality using standard datasets [15, 76, 44, 43] and measure object removal performance [65, 50]. However, to our knowledge, no benchmark explicitly evaluates structural dependencies between objects or validates the correctness of object removal sequences of all objects in the scene.

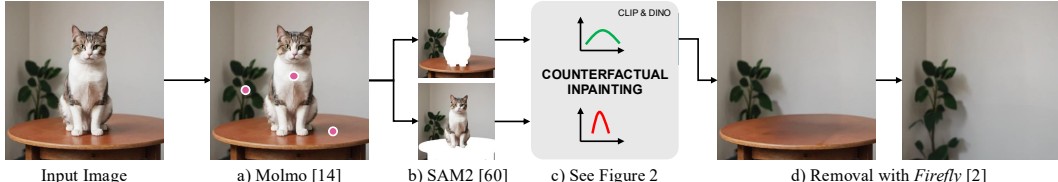

| Input Image | a) Molmo [14] | b) SAM2 [60] | c) See Figure 2 | d) Removal with *Firefly* [2] |

Figure 4: **Our Pipeline.** Given only an input image, (a) we first run Molmo [14] which places a point on each object in the image. (b) These points then serve as prompts for the Segment-Anything (SAM 2) [60] model to obtain segmentation maps for each object. (c) Given the object masks, we can now run our Counterfactual Inpainting method on all object candidates to determine their removal order via a ranking strategy (illustrated in Fig. 2). (d) Finally, we use Firefly [2] to remove objects based on these ranking order.

## 3 Visual Jenga task

Visual Jenga task aims to evaluate scene understanding capabilities beyond passive visual observation, pushing towards physical object interaction understanding [33, 22]. Given a single input image, an algorithm needs to simulate an "action on the scene" by generating a sequence of images where it removes one object at a time until only the background remains while maintaining scene coherency and stability (Fig. 1). Successfully removing objects without destabilizing the scene demonstrates an understanding of object dependencies. We next introduce a simple, training-free approach to the Visual Jenga task that infers removal order based on object co-occurrence, without relying on any explicit physical reasoning.

### 3.1 Dependency as conditional probabilities

Consider the illustrative example on Fig. 3: performing an Internet image search for "cup" returns many images featuring tables, while a search for "table" rarely shows cups. This fundamental asymmetry reveals the dependencies between objects in a scene, and has been used to uncover object causal connections [46]. Let $A$ = cup and $B$ = table. Shown as a Venn diagram in Fig. 3, the asymmetric relationship can be captured by

$$\begin{cases} P(A \mid B) \ll P(B \mid A) & \Rightarrow \text{A depends on B} \\ P(A \mid B) \gg P(B \mid A) & \Rightarrow \text{B depends on A} \end{cases} \tag{1}$$

In our example, $P(A \mid B)$ is very small (i.e., $P(\neg A \mid B)$ is large), while $P(B \mid A)$ is very large (i.e., $P(\neg B \mid A)$ is small). This asymmetric relationship uncovers not only the existence but also the direction of dependency, extending Reichenbach's principle of the common cause [61, 23]. Since our method only replies on object co-occurrences, is is more practical compared to probabilistic theory of causation [29], which requires computing the counterfactual $P(A \mid \neg B)$.

### 3.2 Counterfactual inpainting method

Generalizing to a real scene with more than two objects, the conditional probability $P(A|B)$ is replaced by the complete notation $P(A|B, \text{rest})$, where "rest" denotes the remainder of the scene. To estimate this, we start from the fact that large generative models capture the distribution $P(x)$, where $x$ is an image that may contain objects of any kind. For a particular image $x = X$ that contains objects $A$ and $B$, we can approximate $P(A|B, \text{rest})$ with $P(A|X - A)$ by masking out object $A$ and using a large generative model to inpaint (hole-fill) the region corresponding the mask, given the rest of the image. Similarly, we obtain $P(B|X - B)$ for object $B$. By comparing these two quantities according to the rules in Eq. (1) as illustrated in Fig. 2, we can infer the object dependencies purely from co-occurrence statistics.

### 3.3 Practical details

The above algorithm is simple and principled, but to make it practical, we need to specify how to: (i) obtain object masks, (ii) reliably compute conditional probabilities of inpainted images, and (iii) choose which object to remove first. We describe our choices below and show them in Fig. 4. None of these choices should be considered definitive. We expect them to change as technology matures.

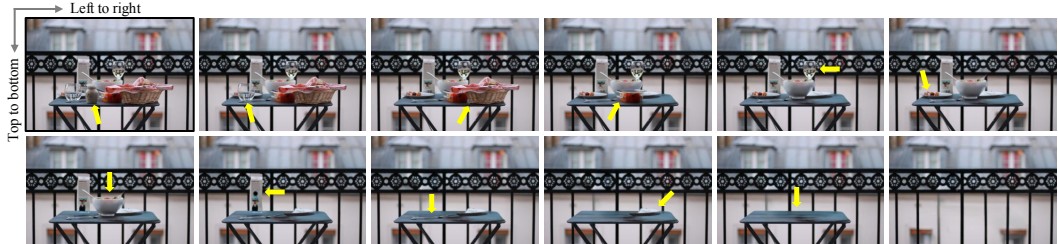

Figure 5: **Generated removal sequence on a breakfast table from our pipeline.** Our method effectively ranks and removes both visible and occluded objects. Our method deals with the occluded plate, introduced after the basket is removed, by rerunning the pipeline again.

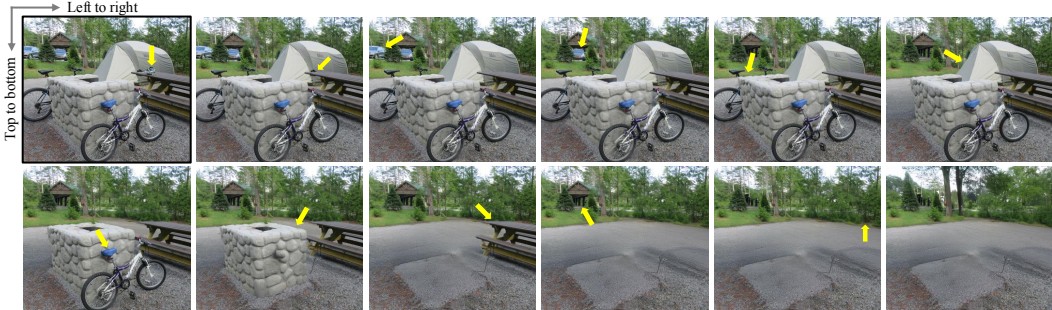

Figure 6: **Generated removal sequence of an outdoor scene from our pipeline.** Smaller objects on the table are cleared before the table itself. The bicycles are taken away before the stone wall, which they are leaning against. Better viewed in our animated video included on our project page.

**(i) Obtaining the object mask.** To get object masks in the scene, we use off-the-shelf models. We first extract object coordinates using MOLMO [14] (Fig. 4a), and then use these as prompts for SAM 2 [60] to obtain segmentation maps without class labels (Fig. 4b).

**(ii). Obtaining reliable conditional probability.** Directly extracting likelihoods from image diffusion models is unreliable for two reasons: first, $P(A \mid X - A)$ for a specific object $A$ is noisy; second, diffusion models are not optimized for likelihood scoring [30, 36]. Therefore, rather than focusing on a specific instance of $A$, we consider a semantic class of $A$ and evaluate the "peakedness" of the distribution $P(A \mid X - A)$. We call this measure the **diversity** score of $A$ (Fig. 4c). To compute it, we first gather $N$ different inpaintings of $A$, denoted $c_{\text{new}}^j$ for $j \in [1, N]$, using Runway's checkpoint of Stable Diffusion 1.5 [64]. Note: the inpainting model matters. We have found that many current inpainting models (such as the newer stable diffusion (XL, 2, 3) [58, 21] seem to be over-reliant on textual inputs. Without clear textual guidance, their inpaint results don't seem to reflect the underlying P(A | rest) of image statistics. For instance, putting a human head on top of the table. Among all models we tried, we found Runway's checkpoint to be the most reliable option, outperforming several others, including the latest FLUX models [39].

We then quantify how semantically diverse these $N$ inpaintings are using both CLIP [59] and DINO [53] features.

$$1 - \frac{1}{N} \sum_{j=1}^{N} \text{CLIPSIM}(c_{\text{new}}^j, c_{\text{orig}}) \times \frac{1}{N} \sum_{j=1}^{N} \text{DINOSIM}(c_{\text{new}}^j, c_{\text{orig}}) \qquad (2)$$

where CLIPSIM and DINOSIM are cosine similarity over CLIP and DINO representations (normalized to $[0, 1]$ and normalized by the segmentation area fraction of the crop), $c_{\text{orig}}$ is the original crop. Using either CLIP or DINO alone works, but having both is more robust (see Table 3).

**(iii) Removing the most "diverse" object.** After computing the diversity scores for all objects in the image, we remove them in order, starting from the *highest* diversity score, using an off-the-shelf object remover, Adobe Firefly [2] (Fig. 4d). If a new object appears (for example because of occlusion) after an object removal, it is included in the ranking by rerunning the whole pipeline again. We found that rerunning the entire pipeline after each removal led to more accurate rankings, particularly in cases where similar or identical objects were stacked on top of each other.

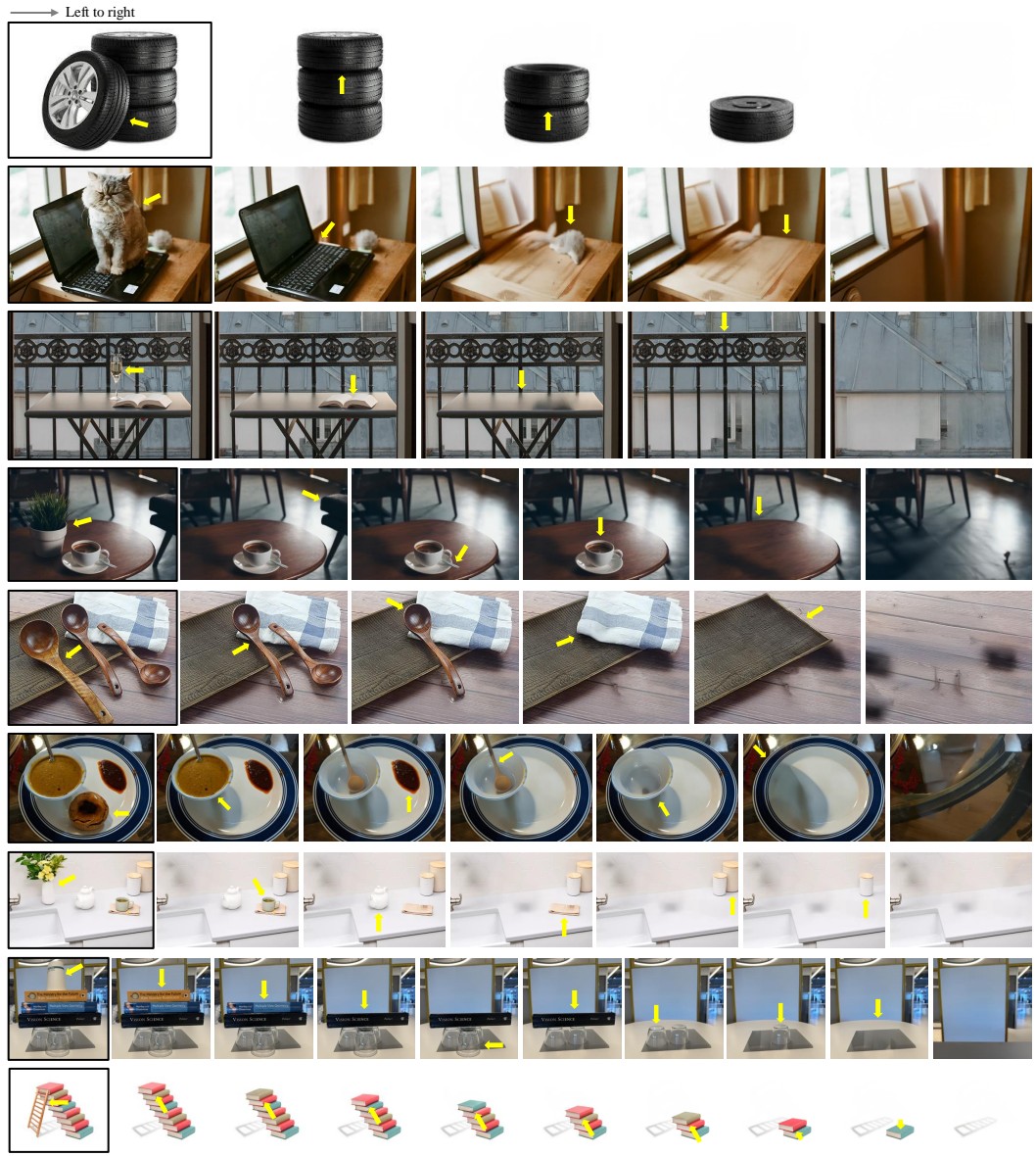

Figure 7: **Generated removal sequences on diverse images with increasing object counts** (top to bottom). Yellow markers indicate the *next* object to be removed. In the second row, the cat is removed first, followed by the laptop and table. In the fifth row, the napkin is removed *after* the three serving spoons. In the sixth row, the order is: hard-wheat rolls (*baati*), lentil soup, sauce, spoon, and finally the plate (note that our method even removes the lentil soup). In the second-last row, while one of the three glasses is removed before the last book, the scene remains plausibly stable.

## 4 Evaluation

Human visual inspection, while qualitative, remains the most natural way to evaluate Visual Jenga. To complement this qualitative assessment, we also perform an automatic quantitative evaluation. Our evaluation comprises three parts: pair-wise object ordering (Sec. 4.1), complete scene decomposition (Sec. 4.2), and comparison to simple heuristics (Sec. 4.3). All evaluation data are provided in Supp.

### 4.1 Pair-wise object ordering

To assess object dependency ordering in an automated manner, we test the model's ability to determine which of two given objects (specified by masks) should be removed first when physical constraints dictate a clear order. We created three test sets:

**NYU-v2:** We use NYU Depth V2 dataset [66], which contains 1449 RGBD images of indoor scenes. Using support relation annotations from Yang et al. [75], we extracted 485 unique images yielding 668 pair-wise comparisons with unambiguous removal ordering (details in Appendix C).

**COCO:** We manually collected 200 random images from COCO dataset [43], which contains diverse everyday scenes with common objects. We selected images that have clear support relationships and good segmentation quality. A complete collection methodology is provided in Appendix B.

**ClutteredParse:** Existing datasets like NYU-v2 and COCO contain limited examples of complex object dependencies (e.g., stacks of objects, hanging/leaning objects). To address this gap, we also created ClutteredParse dataset. Using keywords such as "messy desk," "messy room," and "stacked objects," we curated a test set of 40 challenging object pairs from 40 unique internet images, where human experts provided instance-level segmentations and non-trivial removal ordering.

**Results:** Our method achieves 91.3% accuracy on NYU-v2 (610/668 pairs), 79.5% on COCO (159/200 pairs), and 65% on ClutteredParse (26/40 pairs), where the chance is 50%.

## 4.2 Full Scene Decomposition

While the pair-wise comparison is easy to automate, it doesn't evaluate the full sequential aspect of Visual Jenga. Therefore, we also perform a qualitative evaluation to assess the full sequence decomposition. Given that scene segmentation is underdefined, e.g., segmenting a single piece of paper vs. segmenting a whole pile of papers, we require human evaluators to perform post-hoc assessments for overall physical plausibility and geometric coherence of the image sequence.

Table 1: Full-scene decomposition algorithm evaluation.

| Method | Full Scene Decomposition |
|---|---|
| Top-to-Bottom | 41.1% |
| Small-to-Large | 42.9% |
| Front-to-Back | 8.9% |
| **Ours** | **71.43%** |

To this end, we further collected 56 unique scenes, including both our own photography and internet images, using a similar protocol to ClutteredParse. For each of these scenes, we perform sequential object removal until only the background remains, and the human evaluator scores the whole sequence as "pass" or "fail".

**Results:** Our method achieves 71.43% success (40/56 scenes) on the full-scene decomposition dataset. Qualitatively, Fig. 7 shows varying object counts, while Fig. 5, Fig. 6, and Fig. 9 show complex decompositions — including a breakfast table, an outdoor scene, and an office table, where our method successfully removes all objects (see project page for more results).

## 4.3 Comparison to Figure/Ground heuristics

While there is no existing methods for solving our Visual Jenga task, it is somewhat related to the classic perceptual organization problem of figure/ground assignment (which object is in front of the other). Inspired by the Gestalt principles in psychology, a number of heuristic cues have been proposed to determine figure/ground relationships [55]. Some of the most popular cues include: (1) size – smaller objects are usually in front, (2) convexity – objects with convex masks are more likely to be in front, and (3) surroundedness – an object completely surrounded by another tends to be in front. We have used these cues for comparison, with a simple assumption that the closer object should be removed first. We have also added two other simple cues: (4) front-to-back – using single-view

Table 2: Pair-wise object ordering: algorithm evaluation and baseline comparisons.

| Dataset / Method | Pair-wise Comparison | | |
|---|---|---|---|
| | **NYU-v2 [66]** | **COCO [43]** | **ClutteredParse** |
| Top-to-Bottom | 59.1% | 45.0% | 52.5% |
| Small-to-Large | 90.1% | 74.5% | 50.0% |
| Front-to-Back | 12.6% | 32.5% | 47.5% |
| Convexity | 69.0% | 64.0% | **67.5%** |
| Surroundedness + Top-to-Bottom | 83.1% | 67.5% | 55.0% |
| Surroundedness + Small-to-Large | 90.4% | 74.5% | 50.0% |
| Surroundedness + Front-to-Back | 35.3% | 52.5% | 50.0% |
| Surroundedness + Convexity | 70.8% | 70.0% | **67.5%** |
| **Ours** | **91.3%** | **79.5%** | 65.0% |

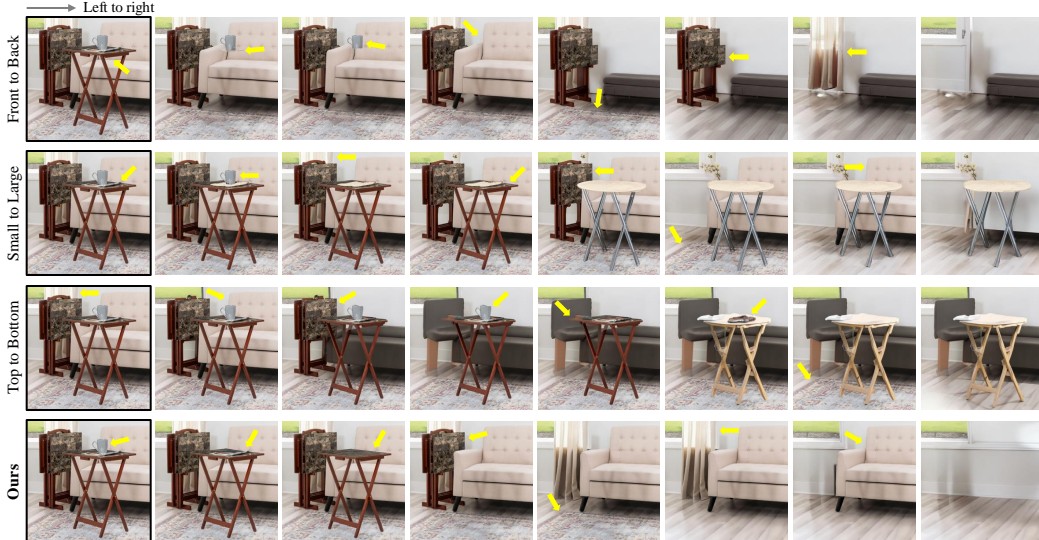

Figure 8: **Comparison against simple heuristics.** Simple heuristics, such as front-to-back ordering (top row; removes table before the cup), small-to-large ordering (second row; removes newspaper before the cup), and top-to-bottom ordering (third row; removes table before the newspaper) can sometimes work, but fail for complex real-world scenes. In contrast, our pipeline (bottom row), based on billion-scale scene statistics, removes objects in a physically and semantically coherent order.

depth estimation [35] algorithm to sort object in depth, and (5) top-to-bottom – the object which is higher in the image (based on its topmost pixel) is removed first.

**Results:** As shown in Table 2 (pair-wise ordering), and also in Table 1 (full-scene decomposition) and qualitatively in Fig. 8, no single heuristic performed well across all datasets. However, two heuristics stood out: small-to-large excelled on simpler scenes (90.1% on NYU-v2, 75.5% on COCO) but struggled with cluttered scenes (50% on ClutteredParse), while convexity, which exploits shape bias, performed consistently across datasets, though less competitively on simpler ones (69.0% on NYU-v2, 64% on COCO, 67.5% on ClutteredParse). The surroundedness cue was generally useful but didn't substantially improve upon the stronger heuristics. In conclusion, while it is possible to pick an appropriate hand-designed heuristic for a given dataset, our data-driven approach performs well across the board.

### 4.4 Ablation studies

**Diversity score.** We compare different ways of quantifying semantic diversity—CLIP, DINO, or both—which are used for measuring the diversity score in Eq. (2). Table 3 shows that using both CLIP and DINO together gives better performance, especially on ClutteredParse—even though it was never used for hyperparameter selection.

Table 3: Diversity score: CLIP vs. DINO.

| Dataset / Method | NYU-v2 | COCO | ClutteredParse |
|---|---|---|---|
| Ours (w/o DINO) | 89.52% | 78.5% | 55% |
| Ours (w/o CLIP) | 90.27% | 78.5% | 57.5% |
| **Ours (full)** | **91.32%** | **79.5%** | **65%** |

**Effect of the number of inpainting samples.** The larger number of inpaintings ($N$) helps better capture the distribution of possible scene completions and monotonically increases the performance, but with diminishing returns beyond $N = 8$ (see Appendix F). We used $N = 16$ by default.

### 4.5 Solving Visual Jenga via text prompting

It is reasonable to ask how well can Visual Jenga be solved with modern Vision-Language Models (VLMs), such as ChatGPT 4o. It's easy to prompt a VLM to generate text instructions of the order in which objects should be removed from the image, but this text-output version of Visual Jenga can overstate the true image understanding of the model. E.g., a generated text description "*remove the book on the table*" requires further interpretation—"*which book?*" "*which table?*"—that only

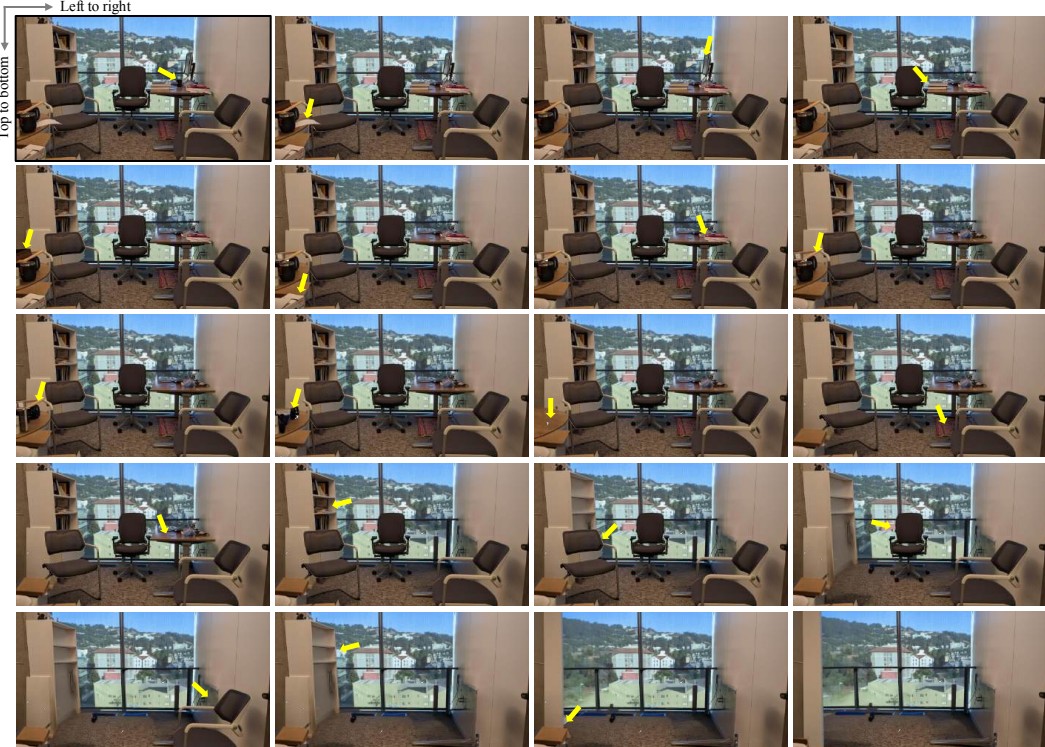

Figure 9: **Emptying an entire office room.** Our method accurately removes all objects from a complex arrangement, reliably removing items such as the carpet mat under the table and books from the bookshelf before removing the bookshelf itself, while preserving physical plausibility. Better viewed in our animated video included in Supp.

becomes clear after demonstrating the full scene decomposition visually. For a more fair comparison, we made VLMs actually generate images of the stages of Visual Jenga, by 1) using native image synthesis capabilities of modern VLMs like ChatGPT 4o, 2) applying text instructions to image editing tools [10], or 3) combining text instructions with our own pipeline (see Appendix G). Despite convincing textual descriptions, the visual results were largely ineffective – the models were either confused about which object was being referenced (Fig. 20), or had trouble preserving original image content (Fig. 18).

Finally, we also experimented with the latest text-prompted video models (Veo 3 [62] and Sora 2 [52]), which were released after our paper was accepted. We were happy to see the models often produce very nice qualitative results on our task (and report about 50% success rate quantitatively on Veo 3 [70]). This agrees with our intuition that temporal data is helpful for scene understanding tasks such as ours. But it's still interesting to see how well our image-only solution can do.

## 5   Limitations

The current approach is slow, requiring multiple inpainting passes per object, and produces only a sequential removal order; unlike humans, who can envision multiple plausible removal strategies simultaneously. Our method relies on Molmo and SAM for object detection and fails when any of these models fail (Fig. 11). Rather than using a pipeline, it might be fruitful to investigate an end-to-end approach, where counterfactual reasoning is used for both, object segmentation [51] as well as scene parsing. Moreover, our approach lacks explicit physical reasoning, relying solely on statistical co-occurrences. It will be valuable in future to build methods that incorporate physical and causal reasoning—such as modeling interventions ($P(A \mid \text{do}(\neg B))$) [57]—potentially by leveraging video generative models [5, 11] as they continue to mature into capable world simulators. Finally, Visual Jenga also raises fundamental questions about object granularity: should an object be defined as a single sheet of paper or as an entire stack? Understanding how generative models internally represent compositional structure could offer deeper insights into object perception and reasoning.

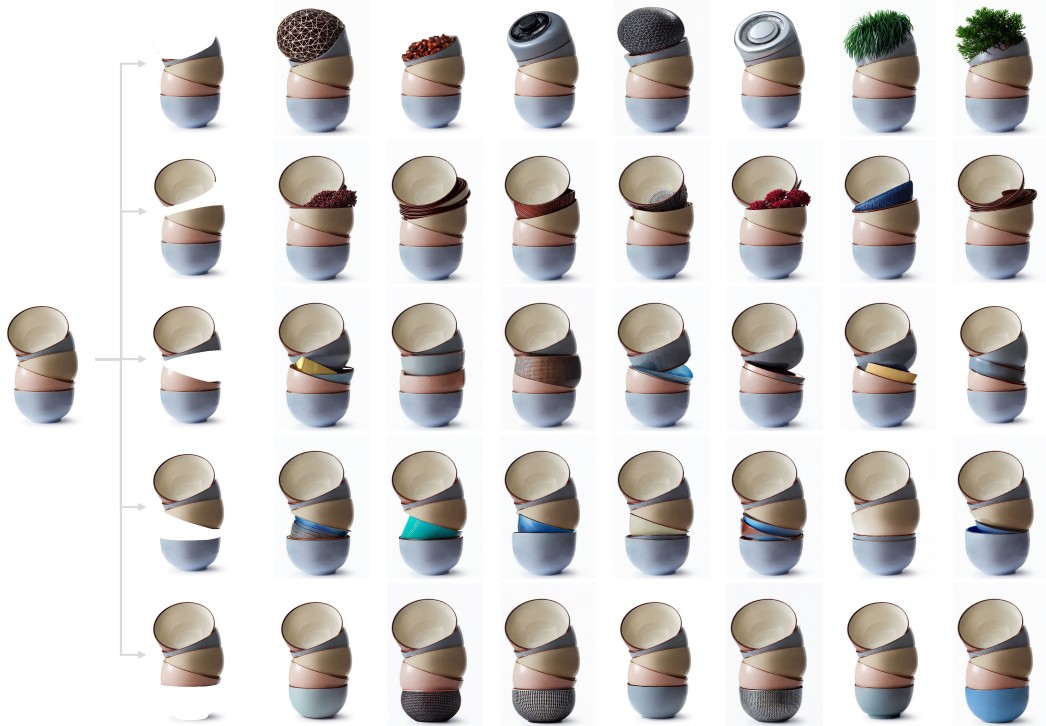

Figure 10: **Inpainting variations on stacked bowls.** Inpainting the top bowl results in a variety of semantics, including fruits, plants, and other objects that happen to fit the second bowl. However, this diversity decreases toward the bottom of the stack, where there are only occasional flowers, a few new bowls added in the middle, and mostly just bowls at the bottom. To reduce ambiguity, rerunning the full pipeline after each removal is suggested; the figure displayed here illustrates only a single pass. After the top bowl is removed, inpainting the new top bowl also produces diverse semantics.

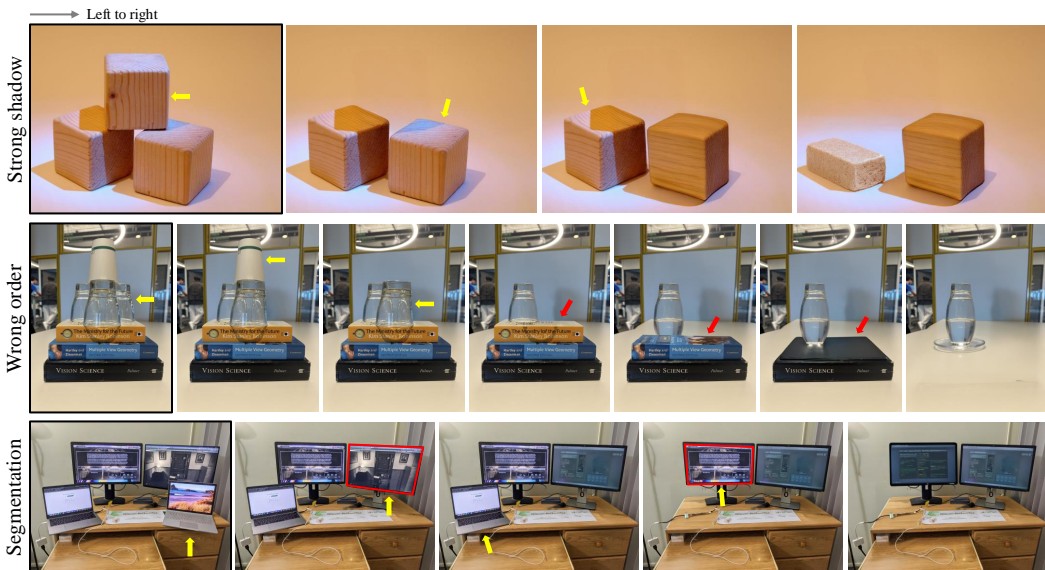

Figure 11: **Failure cases.** Our pipeline can fail for several reasons. *Top Row:* strong shadow cues near an object mislead Firefly, causing it to reinsert a new object rather than remove it. *Middle Row:* our ranking is wrong (indicated by →) because Molmo fails to identify the third glass, leaving it behind. *Bottom Row:* SAM segments only the monitor's screen (indicated by a red polygon) rather than the entire monitor, preventing its removal.

**Acknowledgements:** We are grateful to Yossi Gandelsman and Yutong Bai, whose prior experience with this problem and thoughtful insights have guided our work. We thank Qianqian Wang for her suggestions on our experiments, Amil Dravid and Xiao Zhang for their feedback on the manuscript, and Krishna Kumar Singh for helping us figure out object removal using Firefly. We also thank Leon Bottou for providing the initial inspiration [46] for this project, and for suggesting a connection to Reichenbach's principle [61] and probabilistic causation [29]. This research has been supported in part by ONR MURI grant and NSF IIS-2403305. The work was initiated while AB was a summer visitor at UC Berkeley from TTIC, supported by a grant from TRI.

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

# Appendix

## A    Result compilation video

We provide a result compilation video of Visual Jenga, showcasing the solutions discovered by our proposed method across various scenes in the project page.

## B    Full dataset availability

All images in our evaluation datasets are provided as HTML webpages in the project page for comprehensive inspection. This includes:

1. **Full Scene Decomposition dataset:** 56 scenes collected both from our own photography and from internet searches using keywords such as "messy desk", "messy room", and "stacked objects". For each scene, we perform sequential object removal until only the background remains.

2. **Pair-wise object ordering dataset:**

   - **NYU-v2:** The NYU Depth V2 dataset contains 1449 original images. Using support relation annotations from Yang et al. [75], we extracted 485 unique images yielding 668 pair-wise comparisons with unambiguous removal ordering. Due to the limitation of the class-level (rather than instance-level) support relationship annotations from Yang et al., we carefully filtered the dataset to only include unambiguous cases. The original support label annotations from the NYU Depth V2 dataset are no longer accessible online. Despite our best efforts to contact the original authors and others who had access to the annotations, we were only able to obtain the data with difficulty. Unfortunately, the knowledge required to interpret and utilize these labels has been lost over time. Consequently, we opted to use the alternative annotations provided by Yang et al.

   - **COCO:** We collected 200 images randomly from the COCO dataset (COCO 2017 train split) and used the ground truth instance segmentation that came with the dataset. Our collection methodology was as follows: (1) randomly select an image, (2) retrieve all instance segmentations in the image, (3) keep only segmentations that are not too small (larger than 1% of the total image area), (4) create all possible pairs of segmentations that are spatially next to each other (within 1 pixel radius), and (5) manually review the pairs, keeping only those with clear/unambiguous support relationships and good segmentation quality that covers most of the object area. We repeated this process for 200 images in random order to reduce selection bias. Note that a single image may contain multiple support relationship pairs; however, we tried to avoid reusing the same image multiple times in the dataset, unless the pairs demonstrated different kinds of support relationships that are visually distinct.

   - **ClutteredParse:** Because NYU-v2 dataset has very few examples of complex object dependencies (e.g. stacks of objects, hanging/leaning objects), we produced a more difficult dataset of 40 challenging object pairs from 40 unique internet images. Using keywords such as "messy desk", "messy room", and "stacked objects", we curated this test set where human experts provided instance-level segmentations and non-trivial removal ordering.

As shown in Figure 12, we provide examples from both the NYU-v2 and ClutteredParse pair-wise datasets. In these examples, the model is presented with an image and two segmentation masks (A and B), and must determine which object should be removed first. In both cases shown, object A is correctly identified as the one to remove first.

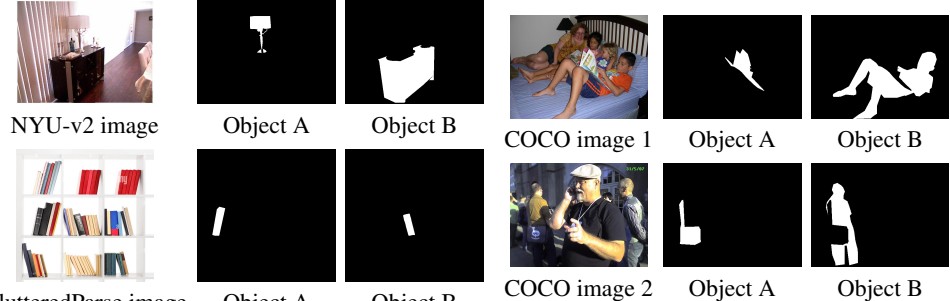

Figure 12: Examples from NYU-v2, ClutteredParse, and COCO pair-wise sets. Left: NYU-v2 and ClutteredParse examples. Right: Two examples from COCO. For each, a model is shown an image and the segmentation masks A and B, and must determine which object should be removed first.

## C   Example of NYU-v2 pair-wise dataset

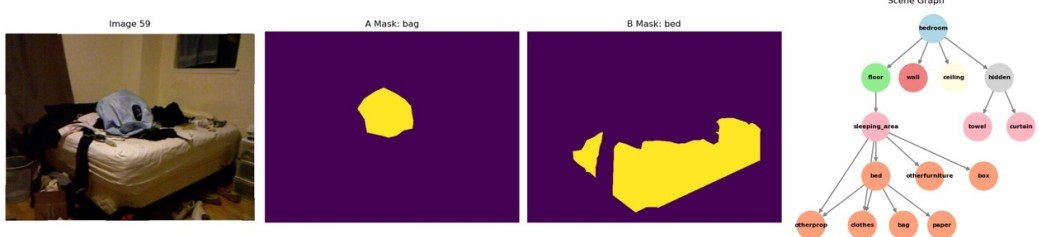

From left to right: (1) Original RGB image showing a bedroom scene, (2) Choice A: segmentation mask for the pillow, (3) Choice B: segmentation mask for the bed, and (4) Scene graph representation showing support relationships obtained from Yang et al. [75]. The scene graph indicates that the pillow is supported by the bed. This example demonstrates how we extract unambiguous pair-wise removal orderings from the annotated support relations in the dataset. Note that the scene graphs from Yang et al. are class-level rather than instance-level annotations, which can be ambiguous in scenes with multiple instances of the same class. We carefully filter out such ambiguous cases and only include examples where the support relationship is unambiguous. Note that the model is provided with only (1), (2), and (3), not the scene graph (4), and must make the decision between removing choice A (pillow) versus choice B (bed) first.

## D   Inpainting details

Since the time of our paper, the original Runway's checkpoint has been deprecated, but there are alternate mirrored third-party versions: https://huggingface.co/stable-diffusion-v1-5/stable-diffusion-inpainting. We also apply a small dilation to the mask before inpainting to minimize shape bias that might arise from the mask itself. Additionally, we observed that inpainting results are more effective when working with images that have a similar aspect ratio, rather than converting them into square images. We make the best efforts to reduce textual biases in a T2I model with a generic prompt "Replace Object, Physically Stable, Realistic, Full HD, 4K, high quality, high resolution, photorealistic". We use the following generic, widely-used, negative prompt: "text, unstable, bad anatomy, bad proportions, blurry, cropped, deformed, disfigured, duplicate, error, extra limbs, gross proportions, jpeg artifacts, long neck, low quality, lowres, malformed, morbid, mutated, mutilated, out of frame, ugly, worst quality. For quantitative scores, all crops are square-shaped, resized to $224 \times 224$ as required by CLIP and DINO, and zero-valued outside the segmentation area. Note that this measure only requires an inpainting model, not necessarily a text-to-image model.

## E   Ablation on the similarity metrics

In addition to the quantitative results provided in Table 3, we also present qualitative ablation for using both CLIP and DINO scores for our ranking. In Fig. 14, we show the removal sequence when

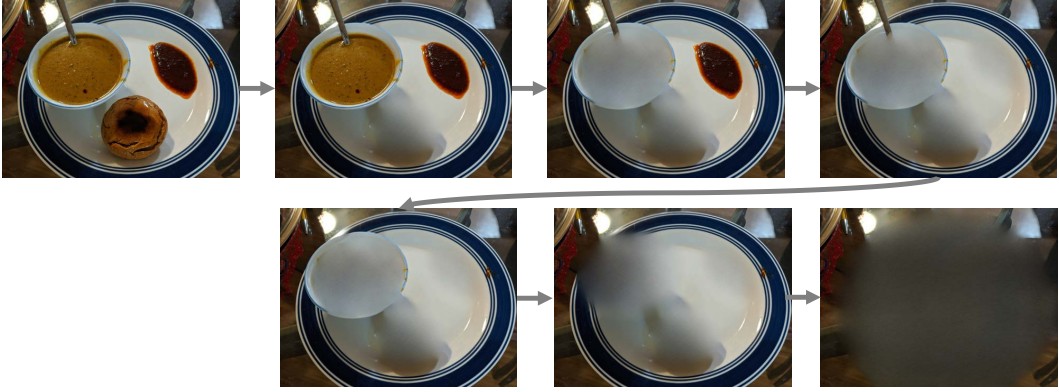

(a) Object Removal Order when emptying a food plate.

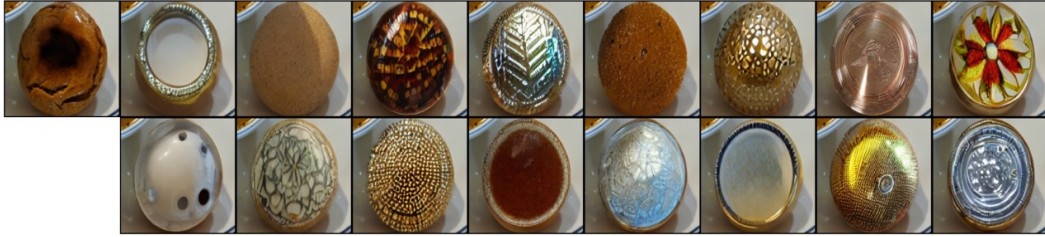

(b) Dumpling: Variance in Dumpling's replaceability is high with many different object types and hence it is removed first.

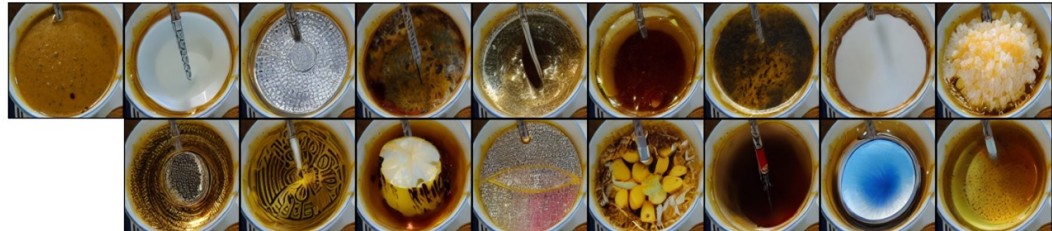

(c) Soup in the bowl: The soup in the bowl can be replaced with many different soups, curd, milk, or other fluid types. But based on our scoring it is the second choice for removing.

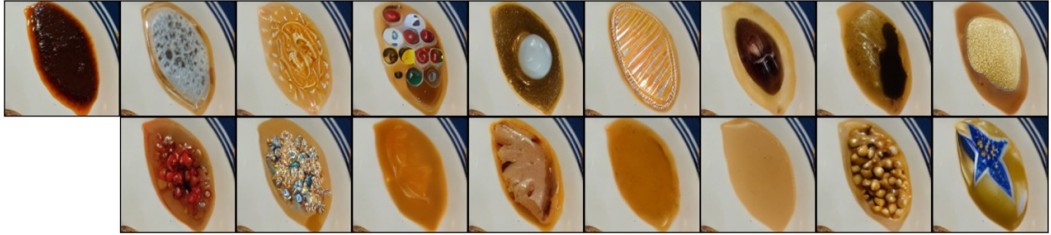

(d) Red Sauce

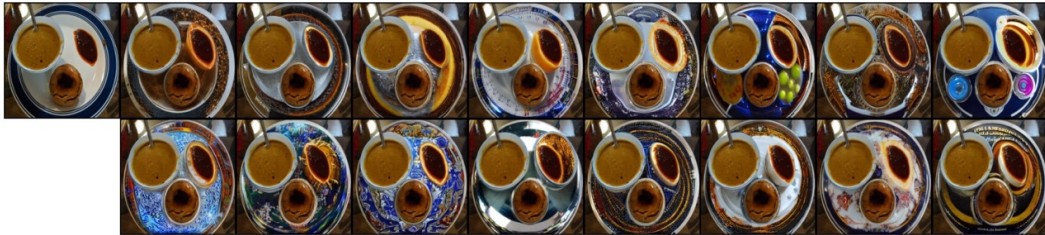

(e) Plate has very limited replacements possible and hence it is the last thing that is removed from the scene.

Figure 13: Visualization of object replaceability through multiple inpainting variations. The original object (a) and three different inpainting results (b-d) demonstrate the range of possible replacements while maintaining scene coherence. Higher visual diversity in replacements indicates greater replaceability.

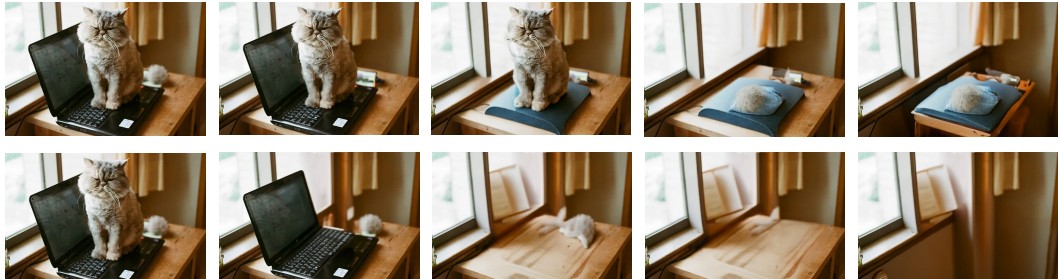

Figure 14: **Ablation: without using CLIP scores.** In the top section, we show the removal sequence without using CLIP scores. In the bottom section, we show the results when both DINO and CLIP scores are used. We observe that DINO tends to favor smaller objects. When CLIP scores are not included, the ordering can be incorrect.

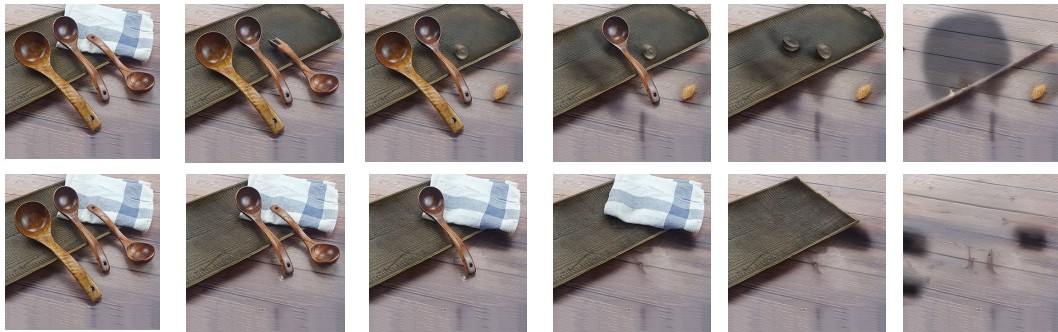

Figure 15: **Ablation: without using DINO scores.** In the top section, we show the removal sequence without using DINO scores. In the bottom section, we show the results when both DINO and CLIP scores are used. We observe that CLIP tends to overlook thin structures. When scoring between the crops, it still recognizes the spoon on top and assigns a high similarity score, which leads to the napkin being removed first.

not using the CLIP scores, and in Fig. 15, we show the removal sequence when not using the DINO scores. The combination of both CLIP and DINO together gives substantially better performance, particularly on ClutteredParse. Since ClutteredParse was used for hyperparameter selection, this suggests that our decision choices are generalizable across scene types.

## F  Ablation on the number of inpainting samples

The larger number of inpaintings ($N$) helps better capture the distribution of possible scene completions and monotonically increases the performance, but with diminishing returns beyond $N = 8$. We used $N = 16$ by default. The performance on ClutteredParse across different values of $N$ is shown in the following table:

| $N = 2$ | $N = 4$ | $N = 8$ | $N = 16$ |
|---------|---------|---------|----------|
| 50% | 50% | 62.5% | 65% |

## G  VLM baselines

We explore VLM-based solutions for Visual Jenga, noting that VLMs don't directly output image sequences. As discussed in the main text, purely text-based solutions risk enabling shortcuts without true image understanding, as text outputs like "remove the book on the table" lack precise object localization and spatial reasoning [72]. We propose integrating ChatGPT 4o (March 2025) as a strong VLM baseline through three pipelines:

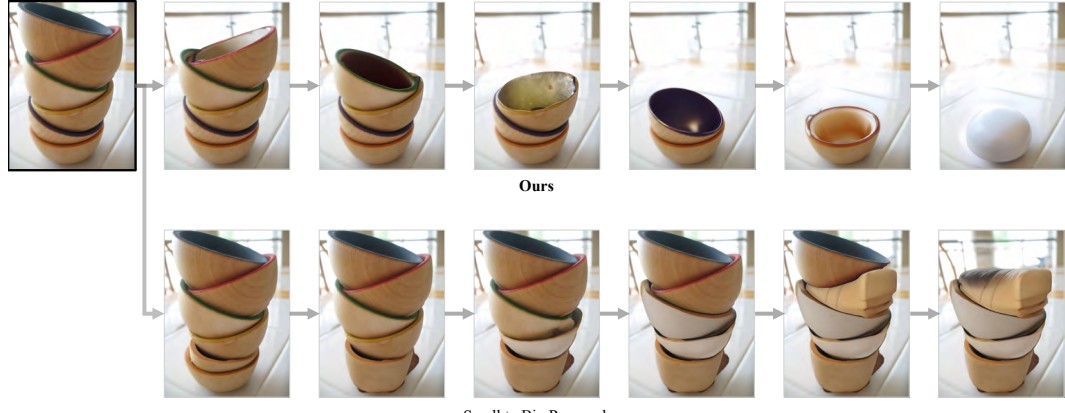

**Ours**

**Small to Big Removal**

Figure 16: **A failure case of a small-to-large heuristic (in the bottom)**. A heuristic approach may work in a few cases but fails in many other cases. Ours (in the top) fail on the last removal because strong shadow force to add objects instead of removing them.

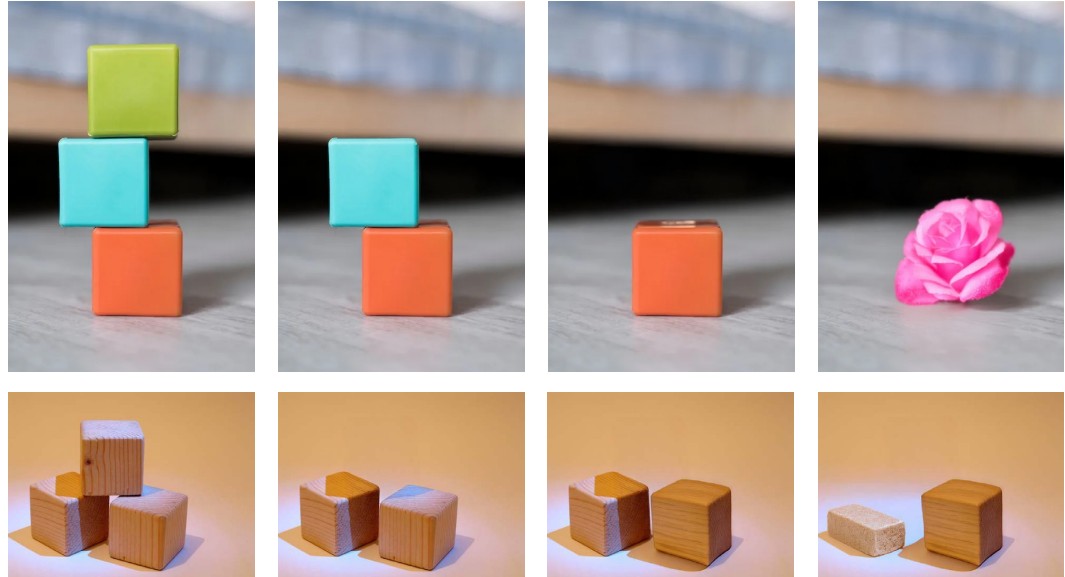

Figure 17: **Failure cases due to (Shadow Residuals).** Even when the sequence is correct, strong shadow cues can lead to incomplete removal. Instead of removing the object, Adobe Firefly responds to these cues and adds a new object instead.

1. **ChatGPT + image generation**: Direct image to image sequence generation as described in Sec. G.1.

2. **ChatGPT + InstructPix2Pix** [10]: Image to image editing with text prompts like "Remove <object>" as described in Sec. G.2.

3. **ChatGPT + Molmo + SAM + Adobe Firefly** [14]: Translates text to visual outputs through object localization and segmentation as described in Sec. G.3, offering a similar pipeline to ours.

Through extensive experimentation with different text prompts, we found that ChatGPT generally identifies correct object removal orders when interpreted by humans. However, these textual descriptions can still be ambiguous when translated into precise spatial locations. The key distinction is that our vision-based approach works directly on object segments, while the VLM methods must first translate textual outputs into spatial locations on the scene.

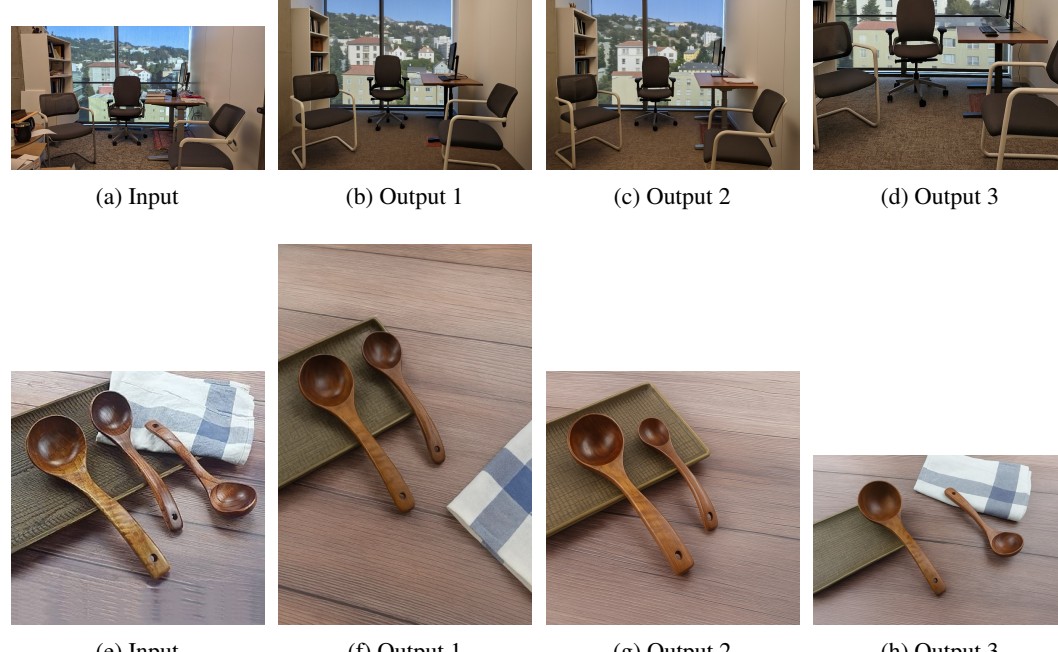

| (a) Input | (b) Output 1 | (c) Output 2 | (d) Output 3 |

| (e) Input | (f) Output 1 | (g) Output 2 | (h) Output 3 |

Figure 18: **ChatGPT + Image Generation [1].** Comparison of the original images and the generated scenes after attempting to remove the first object. The ChatGPT 4o image generation model does not reliably preserve the original image content, making it difficult to determine which objects the model intended to remove.

### G.1 ChatGPT 4o with image generation

ChatGPT 4o's image generation capabilities [1], introduced by OpenAI on March 25, 2025, represent a significant improvement over previous models such as DALL-E. However, as we demonstrate here, its editing abilities on the Visual Jenga tasks still struggle to preserve the original scene content, making it difficult to determine which objects the model intended to remove.

> **ChatGPT Prompt**
>
> "Look at the image I gave. Find object to remove such that after removed the stability of the scene is maintained. The scene is stable if we hypothetically play the scene forward in time, the scene still maintains the same configuration. Then, use your image generation capabilities and give me the image after the object is removed."

See the results in Fig. 18.

### G.2 ChatGPT + InstructPix2Pix

> **ChatGPT prompt:** "Give me the sequence order in which I should remove all objects from the scene until I am left with the background. Don't group objects. Each object should be removed one at a time and shall not disturb the stability of the other objects."
> **InstructPix2Pix prompt:** "Remove <the first object>".

See the results in Fig. 19.

**Input image 1—ChatGPT output:**

For the InstructPix2Pix image, see Fig. 19a.

To remove objects one at a time without disturbing the stability of other objects, here's the suggested sequence of removal:

1. **Papers on the desk**: Start with the loose papers on the desk to reduce clutter.
2. **Mugs on the desk and side table**: Remove the mugs next, as they are small and easy to move.
3. **Glasses on the desk**: Take the glasses off the desk.
4. **Books and files on the desk**: Remove these one by one carefully.
5. **Small items (e.g., pens or small electronics)** on the desk.
6. **Loose items on the bookshelf**: Clear the items on the shelves, starting from the top shelf to the bottom shelf, to maintain balance.
7. **Chair near the desk (right-hand side)**: Move the smaller chair beside the desk.
8. **Chair near the side table (left-hand side)**: Remove the other smaller chair.
9. **Central chair**: Move the office chair in the middle.
10. **Rug under the desk**: Carefully lift and remove the rug.
11. **Side table**: Remove the side table after clearing all items on it.
12. **Desk**: Remove the desk after ensuring all items on and around it are gone.
13. **Bookshelf**: Carefully move the bookshelf after everything else is cleared.

By following this order, you'll remove objects without compromising the stability of others and leave only the background view.

**Input image 2—ChatGPT output:**

For the InstructPix2Pix image, see Fig. 19b.

To remove all objects from the scene without disturbing the stability of the other objects, follow this sequence:

1. **Remove the paper cup** (topmost object). This object is light and does not support any other objects.
2. **Remove the top glass** from the stack of glasses. Carefully lift the glass without disturbing the ones below it.
3. **Remove the second glass** from the stack.
4. **Remove the third glass** from the stack.
5. **Remove the fourth glass** (bottom-most glass in the stack).
6. **Remove the book "The Ministry for the Future"** (yellow cover).
7. **Remove the book "Multiple View Geometry"** (blue cover).
8. **Remove the book "Vision Science"** (black cover).

This sequence ensures stability as you work your way from the topmost object to the base.

**Input image 3—ChatGPT output:**

For the InstructPix2Pix image, see Fig. 19c.

To remove objects one at a time while ensuring stability, follow this sequence:

1. **Smallest spoon on the right** - Start with the smallest object, as it does not provide structural support to other objects.
2. **Medium-sized spoon in the middle** - Next, remove the medium-sized spoon, as it is similarly isolated and won't destabilize anything.
3. **Largest spoon on the left** - Then remove the largest spoon, as it is freestanding and can be removed without affecting the others.
4. **Napkin on the top right** - Finally, remove the napkin, as it is flat and won't interfere with the background.

5. **Tray on the bottom left** - End with the tray, as it forms part of the background base and is the largest object remaining.

After following this order, you will be left with only the background.

**Input image 4—ChatGPT output:**

For the InstructPix2Pix image, see Fig. 19d.

Here is the sequence of removal to ensure the stability of objects and to avoid disturbing others:

1. **Knife** (on the cake with sprinkles) - It's small and easy to remove without affecting stability.
2. **Can of whipped cream** (right side) - Freestanding and easy to pick up.
3. **Pink drink in the glass** (right side) - Remove carefully to avoid any spills.
4. **Fruit plate** (back right, with kiwi and berries) - This is stable and can be picked up without disturbing others.
5. **Bowl with crumble** (back middle, near the pie) - A lightweight item that is independent.
6. **Bowl of chocolates** (front right) - Remove carefully as it doesn't seem to be stacked or linked to others.
7. **Pie** (front right in a red dish) - Freestanding and can be removed next without disturbing others.
8. **Cheesecake** (middle back, with red topping) - Positioned safely on its plate, making it stable to remove next.
9. **Cake with sprinkles** (front left) - Finally remove this large, stable item.

Now you should be left with just the background.

### G.3 ChatGPT + Molmo + SAM + Adobe Firefly

> **ChatGPT Prompt**
>
> "Give me the sequence order in which I should remove all objects from the scene until I am left with the background. Don't group objects. Each object should be removed one at a time and shall not disturb the stability of the other objects."

This pipeline is described as follows: First, obtain the removal order from ChatGPT using the prompt from Sec. G.2 above. Second, translate the textual removal order into image segmentation using Molmo to point given the object's text description, and then SAM to segment from a point. Third, remove each object in order using Adobe Firefly given the object's segmentation. These steps are important to turn textual output from ChatGPT into a visual output expected by Visual Jenga.

This solution can solve all simple cases. However, failure cases in Fig. 21 suggest that: First, the textual description of the object and its location can be ambiguous which leads to pointing errors from Molmo. This kind of error is more likely to happen in a scene with multiple objects of the same kind as also observed in Fig. 20 (b). Second, ChatGPT does make mistakes. Finally, we show side by side comparison with our proposed method showing different modes of failures between the two methods in Fig. 20: our proposed method may suffer from transparent objects, the ChatGPT method may suffer from ambiguous object descriptions.

**Input image 1—ChatGPT output:**

For the image sequence, see Fig. 20b (top).

To remove all objects from the scene without disturbing stability, follow this order:

1. Remove the **top paper cup** gently.
2. Remove the **book "The Ministry for the Future"** (orange book).
3. Remove the **book "Multiple View Geometry"** (blue book).

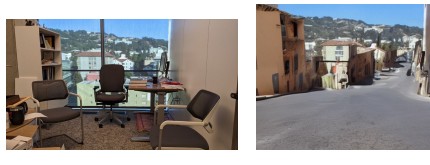

**Input image**      **InstructPix2Pix output**

(a) **ChatGPT output.** To remove objects one at a time without disturbing the stability of other objects, here's the suggested sequence of removal: 1. Papers on the desk...

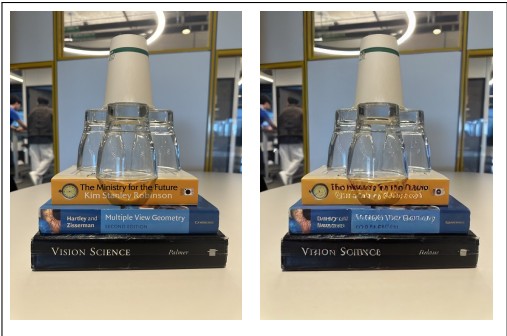

**Input image**      **InstructPix2Pix output**

(b) **ChatGPT output.** To remove all objects from the scene without disturbing the stability of other objects, follow this sequence: 1. Remove the paper cup (topmost object)...

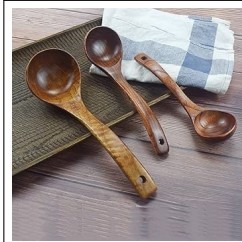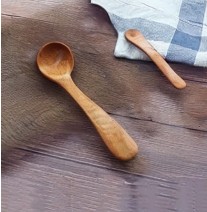

**Input image**      **InstructPix2Pix output**

(c) **ChatGPT output.** To remove objects one at a time while ensuring stability, follow this sequence: 1. Smallest spoon on the right...

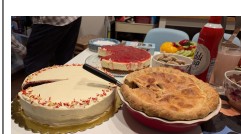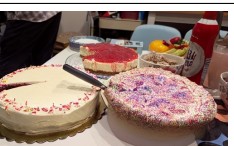

**Input image**      **InstructPix2Pix output**

(d) **ChatGPT output.** Here is the sequence of removal to ensure the stability of objects and avoid disturbing others: 1. Knife (on the cake with sprinkles)...

Figure 19: **ChatGPT + InstructPix2Pix.** Comparison of the original images and updated scenes with the first object removed. Each subfigure is boxed for clarity. InstructPix2Pix cannot follow the prompt to remove an object in the image well.

 4. Remove the **book "Vision Science"** (black book).
 5. Sequentially remove each **glass cup** one at a time from the stack (there appear to be four glass cups, so remove them one by one).

This sequence ensures the stability of the objects as you remove them.

**Input image 2—ChatGPT output:**

For the image sequence, see Fig. 20b (bottom).

To remove all objects from the scene without disturbing the stability of the other objects, follow this sequence:

 1. Remove the paper cup (topmost object).
 2. Remove the top glass from the stack of glasses.
 3. Remove the second glass from the stack.
 4. Remove the third glass from the stack.
 5. Remove the fourth glass (bottom-most glass in the stack).
 6. Remove the book "The Ministry for the Future" (yellow cover).
 7. Remove the book "Multiple View Geometry" (blue cover).
 8. Remove the book "Vision Science" (black cover).

By following this order, you'll remove objects without compromising the stability of others and leave only the background view.

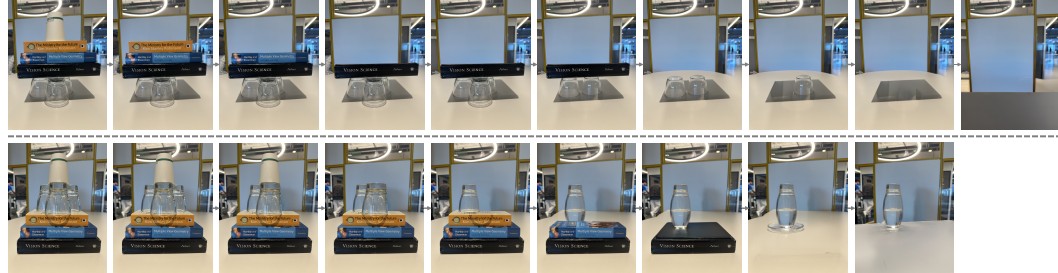

(a) **Solutions from our proposed counterfactual inpainting pipeline.**

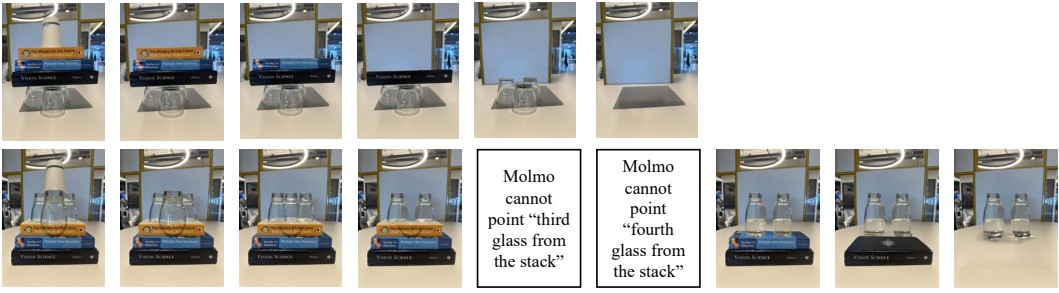

(b) **Solutions from using VLM (ChatGPT 4o) with a similar pipeline (Molmo + SAM + Firefly).** For the top row, at the last step, ChatGPT suggests "Sequentially remove each glass cup one at a time from the stack (there appear to be four glass cups, so remove them one by one)", which Molmo points to all of the glass cups effectively removing them all at once. For the second row, after removing the paper cup, ChatGPT suggests: "Remove the top glass from the stack of glasses", "Remove the second glass from the stack", "Remove the third glass from the stack", and "Remove the fourth glass (bottom-most glass in the stack)". Such description can be ambiguous, and Molmo failed to locate the "third" and the "fourth" glasses as shown above.

Figure 20: **Comparing our counterfactual inpainting with a VLM-based method (ChatGPT 4o) sharing a similar pipeline.** demonstrates different failure modes from the two approaches. The ChatGPT solution has a bottleneck in coming up with a clear text description for locating an object which is exacerbated where there are many similar objects, e.g. glasses, in the scene.

**Input image 3—ChatGPT output:**

For the image sequence, see Fig. 21 (top).

> In the image, there are four stacked tires and a wheel. To remove all the objects one at a time without disturbing the stability of the remaining objects, follow this sequence:
>
> 1. **Top tire** - Remove the tire at the very top of the stack first.
> 2. **Second tire** - Remove the tire directly beneath the top tire.
> 3. **Third tire** - Remove the next tire in the stack.
> 4. **Bottom tire** - Remove the last tire in the stack.
> 5. **Wheel** - Finally, remove the wheel, as it is resting separately and won't affect the stability of any other objects.
>
> This sequence ensures that the stack's stability is maintained during the removal process.

**Input image 4—ChatGPT output:**

For the image sequence, see Fig. 21 (bottom).

> To ensure stability while removing the objects one at a time:
>
> 1. Remove the **small cylindrical object** (bottom right) as it is not supporting any other object.

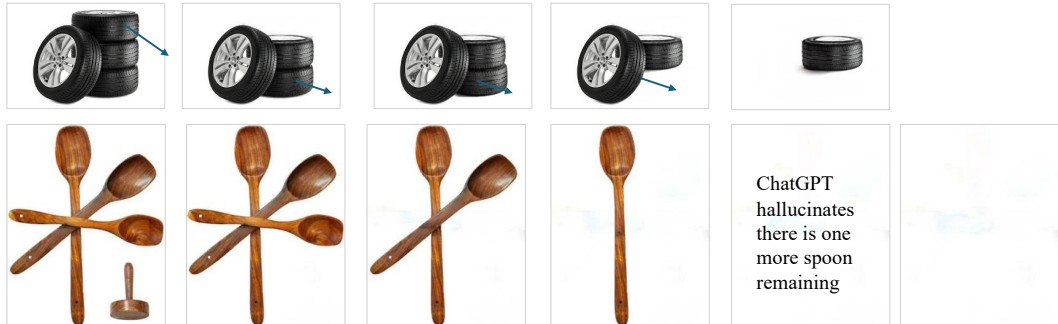

Figure 21: **Failure Cases from ChatGPT + Molmo + SAM + Adobe Firefly.** In the top row, ChatGPT recommends removing the leaning tire last. Additionally, based on its description, Molmo points to the bottom tire twice, likely due to the word "beneath." In the bottom row, ChatGPT mistakenly adds an extra spoon, while Molmo indicates the top spoon in the stack. Below are the output suggestions from ChatGPT for the top row."In the image, there are four stacked tires and a wheel. To remove all the objects one at a time without disturbing the stability of the remaining objects, follow this sequence: 1 **Top tire**—Remove the tire at the very top of the stack first. 2. **Second tire**—Remove the tire directly beneath the top tire. 3. **Third tire**—Remove the next tire in the stack. 4. **Bottom tire**—Remove the last tire in the stack. 5. **Wheel**— Finally, remove the wheel, as it is resting separately and won't affect the stability of any other objects. This sequence ensures that the stack's stability is maintained during the removal process." Below are the output suggestions from ChatGPT for the bottom row. "To ensure stability while removing the objects one at a time: 1. Remove the small cylindrical object (bottom right) as it is not supporting any other object. 2. Remove the wooden spoon leaning at the bottom right diagonally since it overlaps with the others and is not directly supporting them. 3. Remove the wooden spoon at the bottom left as it supports the least weight in the arrangement. 4. Remove the spoon on the top left that overlaps minimally with the one left behind. 5. Finally, remove the last spoon resting horizontally across the center. This sequence ensures stability and minimizes disturbance to the other objects."

2. Remove the **wooden spoon leaning at the bottom right diagonally** since it overlaps with the others and is not directly supporting them.
3. Remove the **wooden spoon at the bottom left** as it supports the least weight in the arrangement.
4. Remove the **spoon on the top left** that overlaps minimally with the one left behind.
5. Finally, remove the **last spoon resting horizontally across the center**.

This sequence ensures stability and minimizes disturbance to the other objects.

## H   Error accumulation from multi-step solutions

Both our proposed solution and the ChatGPT solution in Sec. G.3 involve multiple steps. Each introduces its own kind of error which accumulates. At the segmentation stage, given just a point, segmentation is very much an underdetermined task, and SAM, the state-of-the-art segmentation method, is likely to make mistakes. At the object removal stage, some hard cases involve strong shadows or reflections which are not considered part of the segmentation in a traditional sense but are quite important for correct removal as a strong cue gives away the presence of the object which makes the object removal very hard or impossible. For these reasons, an end-to-end vision-based solution is highly desirable and presents a promising direction for future work.

