# OpenReview forum: "Visual Jenga: Discovering Object Dependencies via Counterfactual Inpainting"
_NeurIPS.cc/2025/Conference — NeurIPS 2025 poster_

### Official Review · Reviewer_r78o · 2025-06-03

[review text omitted: it was posted to a different submission]

---

> ### Author Rebuttal · Authors · 2025-07-31
>
> We thank you for the excellent summary and constructive questions, and for recognizing our task as "original and well-motivated" and our method as "creative." In general, we are thrilled that the reviewers unanimously found the task to be original and significant.
>
> **“The method approximates physical dependency via visual diversity, which is not equivalent to support or causality. There is no explicit modeling of physical forces, stability, or geometry.”**
>
> Yes, there is no explicit modeling of physical forces, stability or geometry in our method. But given only a single input image, it is impossible to measure such information. One could try to approximate/hallucinate it using learning-based methods, and then apply physical simulation methods, e.g. \[46,71\]. However, to the best of our knowledge, these methods have not yet been shown to perform well on real-world scenes.
>
> Instead, in this paper, we take a purely statistical approach to the problem, studying how far we can get by a correlation-based approach, inspired by \[44\]. Note that we are careful in the paper not to overclaim this as "causal reasoning." In our Related Work section, we explicitly state that our approach is based on co-occurrences and is "not truly causal".
>
> **“The diversity score is heuristic in nature. While it performs well empirically, it lacks formal justification or analysis beyond intuition."**
>
> The diversity score is a heuristic approximation of asymmetric relationships that has a formal basis—Reichenbach’s principle of the common cause \[57, 22\], as discussed in Section 3.1.  Reichenbach's principle links correlation to a common cause. Our method takes this further by comparing conditional probabilities (P(A|B) vs P(B|A)) to infer directional dependencies \[44\]. For example, if observing a cup strongly implies a table, but not vice versa, we can infer that the cup is dependent on the table.
>
> Therefore, the diversity score ideally should be the likelihoods (P(A|B) vs P(B|A)).  Unfortunately, contemporary diffusion models do not provide reliable likelihood estimates \[35\] (Sec. 3.3). This motivated our use of CLIP and DINO to create an approximate likelihood estimate that is also practical, as empirically justified in our ablations (Sec. 4.4).
>
> **“Would a learned module (e.g., a small model trained on support relations) outperform the current heuristic-based approach?”**
>
> If we understand correctly, the suggestion is to use our pairwise object ordering test set to train a simple pairwise binary classifier. We have not tried this, but we suspect that our pairwise test set won’t provide sufficient quantity or diversity of data to train a good model, even if coupled with a strong pre-trained vision backbone.  Additionally, it is not immediately clear how to adapt this model for the full-scene Visual Jenga task. We note that our focus in this work is to see how far we can go with a simple, training-free correlation-based approach.
>
> **“Could the method be extended to support alternative removal orders or multiple valid sequences?”**
>
> This is an excellent conceptual question.  Because the task can have multiple valid solutions, a key limitation of our current deterministic method is that it produces only a single removal sequence.  One possible way to address this would be to group similar diversity scores into equivalence clusters, which represent a partial ordering. However, finding the right setting of hyperparameters, e.g., the number of clusters, would be a challenge.
>
> **“How sensitive is the dependency estimation to segmentation errors or missed objects (e.g., occlusions)?”**
>
> The performance is indeed sensitive to errors from the upstream models. We demonstrate these failure modes in Figure 10, which shows examples where an incorrect ranking results from Molmo failing to detect an object, or where a removal is prevented because SAM produces an incomplete segmentation. Error propagation is a known limitation of such pipeline-based systems.
>
> **“Evaluation on full-scene removal relies on human judgment without an objective definition of “scene plausibility”.**
>
> Yes, single image scene plausibility is necessarily subjective. However, determining whether a scene is well-formed is known to be a reasonably unambiguous task for humans \[9\].
>
> **“The scope is limited to static 2D images. The method does not generalize to video, 3D, or interactive settings, where object dependencies are more dynamic and informative.”**
>
> Visual Jenga is a single-image scene understanding task, in the tradition of single view image understanding in computer vision (e.g. \[30,31,33,70\]) and psychology (e.g. \[9\]).  Our goal is to investigate the limits of what can be inferred from a single static image without the benefits of information such as scene geometry and physical properties, or motion, similar to how humans understand still photographs. Just like with humans, this understanding will never be perfect.
>
> In this paper, we propose a simple, correlation-based baseline to tackle the Visual Jenga problem. The proposed method performs surprisingly well, as demonstrated by the human evaluation on full-scene decompositions. Of course, we hope that the subsequent works will explore other, more sophisticated methods, including video and interactive settings, i.e., a “Video Jenga” task—where a dynamic scene is observed before object removal.
>
> **“Have you considered using explicit 3D reasoning or physical simulation to validate or improve the ranking?”**
>
> We did not consider this because, to the best of our knowledge, current methods for performing physical simulations given visual data (e.g. \[46,71\]) are not yet able to work on real-world images, as they are often designed for simplified or synthetic environments.
>
> **“how were human labels collected? Were annotators consistent on what counts as a “stable” scene?”**
>
> As detailed in Supplement Section B, the dataset was curated by a small team of "human experts" (the paper's authors) who provided instance-level segmentations and established the ground-truth removal ordering through discussion and consensus. This ensured a consistent and high-quality, albeit small, benchmark for challenging scenes.

---

> ### Comment · Reviewer_r78o · 2025-08-05
>
> Thanks for your clarification. The authors have adequately addressed my concerns. I have increased my rating and recommend acceptance.

---

> > ### Author Response · Authors · 2025-08-05
> >
> > Thank you for reviewing our rebuttal and for your updated recommendation. We’re glad to hear about the increased rating.

---

### Official Review · Reviewer_1oV5 · 2025-07-03

**Clarity:** 4
**Significance:** 3
**Originality:** 4
**Rating:** 5
**Confidence:** 4

**Summary:**

The paper proposes an interesting new visual task for "visual jenga"; where given a scene with multiple objects, the task is to remove them one by one in a structurally feasible manner until only background remains. For instance consider, an image with a book lying on top of a table in an otherwise empty room. The goal of the visual jenga is to remove the objects one by one in a physically plausible manner. In this case, remove book first and table second. The final image would be of an empty room. The key idea testing on this task helps evaluates the models understanding for structural interdependencies between objects. The paper also proposes a counterfactual inpainting based baseline which removes the object which if inpainting leads to on average the highest CLIP/DINO similarity with the original image. The authors also motivate the need for this task from the perspective of 3D understanding and both qualitative and quantitive evaluations for the proposed approach.

**Questions:**

* The current evaluations in Sec. 4, use other simple object removal ordering as baselines. Can the authors provide evaluations using state-of-art VLMs for these tasks?
* In particular, it would be good to follow the numeric labelling strategy for getting VLM responses to avoid the problem of ambiguity when indicating target object by the VLM
* Also for the curated data, it would be nice to estimate the performance of a simple VLM + posttraining e..g simple SFT. This will provide an estimate of the actual difficulty of the given task, and whether the same can be addressed using more targeted fine-tuning data alone.
* Finally, I was curious about some qualitative examples since their could be multiple orderings for the same scene. For instance, for Fig. 6, I had imagine after removing the cars in the background, removing the hut in the background would lead to the highest DINO similarity between original image and inpainted one. Given the same, it was surprising that the hut was not removed until the very end. Can the authors please provide some explanation for the same?

**Ethical Concerns:**

["NO or VERY MINOR ethics concerns only"]

**Final Justification:**

The author response has adequately addressed my concerns. I therefore increase my score for recommending acceptance.

**Limitations:**

yes

**Quality:**

3

**Strengths And Weaknesses:**

Strengths:
* The proposed task is novel and interesting
* The visual jenga task provides a new way of thinking about how to evaluate current VLM models on their ability to understand 3D scenes and their structural interdependencies.
* The use of counterfactual inpainting is also interesting
* The paper provides both qualitative and quantitative results for showing the efficacy of the proposed approach.

Weaknesses:
* The paper motivates the need for new task for evaluating understanding capabilities of current VLM models. Given the same, it would be good to motivate the need for the same by evaluating current VLM models on this task
* For instance, similar to the proposed pipeline (Fig. 4), the authors could use Molmo for placing a point on each object (with numbering). The generated image could then be fed to a VLM which can respond with which object to remove next.
* Also Sec. 5, the authors mention that text alone as outputs for VLM can lead to ambiguity e.g., if there are multiple objects of same type. However, by placing specific counts on the point for each object (as suggested above), the VLMs should be able to output the desired object to remove without ambiguity.
* Finally, it would be nice to see some comparisons with VLM + SFT (with curated data) to see if the problem can be simply address using targeted data for this task.

---

> ### Author Rebuttal · Authors · 2025-07-31
>
> We thank Reviewer 1oV5 for finding that this work “provides a new way of thinking”, “novel and interesting” task, other helpful comments, and a supportive rating. In general, we are thrilled that the reviewers unanimously found the task to be novel and significant.
>
> We now address the reviewer's concerns and questions.
>
> **“Can the authors provide evaluations using state-of-art VLMs for these tasks?”**
>
> We respectfully point the reviewer to Section G (pages 4-11) of our submitted supplementary material. We conducted this exact line of experimentation using a pipeline nearly identical to the reviewer’s proposal (VLM → Molmo → SAM → Removal), in addition to two more VLM-based pipelines (with 4o Image Generation, with InstructPix2Pix editing), which serve as reasonable adaptations of VLMs for Visual Jenga.
>
> Across all three pipelines, we found a consistent and critical limitation: VLMs, while often conceptually correct in their textual output, struggle with the precise spatial grounding required for a fine-grained task like Visual Jenga. As we demonstrate in Supp. Figures 19 and 20, this becomes particularly evident in cluttered scenes.
>
> This finding highlights the main advantage of our proposed method. By operating directly on visual segments, our counterfactual inpainting approach bypasses this fragile text-to-vision grounding problem entirely. It reasons about the scene's structure visually, not linguistically.
>
> **“it would be good to follow the numeric labelling strategy for getting VLM responses to avoid the problem of ambiguity when indicating the target object by the VLM”**
>
> The reviewer's suggested pipeline may solve the problem of output ambiguity (i.e., making the VLM's response precise). However, our findings suggest a deeper issue may lie in the VLM's core physical reasoning from a single 2D image. For example, our experiments showed VLMs do hallucinate, such as suggesting a _physically incorrect removal order for a stack of tires_ or hallucinating an extra object in a scene **(Supp, Figure 20)**. This suggests the issue is not just in communicating the answer, but in the reasoning process itself.
>
> This highlights the difference in approach. Our method does not rely on the VLM's high-level, and sometimes ungrounded, reasoning. Instead, it uses a more direct visual signal—the diversity of plausible counterfactuals—which our experiments show is a robust proxy for physical dependency.
>
> **“it would be nice to estimate the performance of a simple VLM + posttraining e..g simple SFT. This will provide an estimate of the actual difficulty of the given task, and whether the same can be addressed using more targeted fine-tuning data alone.”**
>
> We assume SFT here refers to Supervised Fine-Tuning. Please note that our paper provides only a small testing dataset for evaluation. There are currently no available datasets for training a Visual Jenga model.  Furthermore, there is no economical way to create a high-quality dataset – capturing scene removal sequences or labeling support relations at scale is a huge undertaking. Therefore, here we focus on zero-shot methods.
>
> **An explanation for the qualitative result in Fig. 6, where the hut is removed last.**
>
> Yes, when removing the hut and inpainting, it would likely produce more vegetation rather than a hut (meaning low DINO similarity), and should be safe to remove early. However, the other objects in the scene are even more replaceable, such as the bicycle and the tent, with many other diverse objects, i.e., even lower DINO similarity. The diversity score is relative, not an absolute value. But of course, as the reviewer mentioned, there are many valid removal orderings.

---

### Official Review · Reviewer_Z1Mo · 2025-07-03

**Clarity:** 3
**Significance:** 3
**Originality:** 3
**Rating:** 5
**Confidence:** 4

**Summary:**

The paper proposes **Visual Jenga**, a scene-understanding challenge where the model must successively “pull” objects from a single image while ensuring the scene still looks physically plausible. The authors:

* Define the task and release benchmark splits based on COCO, NYU-v2, and a new ClutteredParse set.
* Offer a training-free baseline: segment objects → inpaint each one with a diffusion model → rank objects by the diversity of plausible inpaintings (measured with CLIP/DINO). High diversity signals the object is not structurally critical.
* Show this diversity ranking outperforms size, depth, and similar heuristics in pairwise dependency tests and full scene deconstructions.

Overall, the work introduces a fresh probe of object-level dependencies and demonstrates that generative inpainting can act as a zero-shot cue for physical importance.

**Questions:**

1. **Physical validity** – Have you verified that high-score removals are truly stable (e.g., human ratings or physics-engine tests)? Evidence here would solidify the task’s credibility.

2. **Dataset realism** – Any plan to add images or videos with denser, load-bearing stacks? Even a small pilot set would show the benchmark’s growth path.

3. **Backbone sensitivity** – How does performance change with a different diffusion model or feature encoder? One ablation would clarify robustness.

4. **Stronger baselines & runtime** – Have you tried a learned support-graph model, and what is the per-image compute time? These numbers matter for judging practical impact.

**Ethical Concerns:**

["NO or VERY MINOR ethics concerns only"]

**Limitations:**

The paper briefly notes computational cost but skips two key aspects:

Physical-validity caveat – it should acknowledge that diffusion-based “plausibility” may mask structurally impossible scenes and propose human or physics-based verification.

Societal impact – manipulating images to hide structural flaws could enable misleading visuals (e.g., safety or real-estate fraud). The authors should discuss safeguards and ethical use guidelines.

**Quality:**

3

**Strengths And Weaknesses:**

**Strengths**

*Originality & Significance.*  Recasting scene understanding as “Visual Jenga”—remove objects one-by-one while the image remains physically plausible—is genuinely novel.  It pushes beyond familiar segmentation or affordance tests toward causal reasoning: which item is truly load-bearing, which is cosmetic?  That framing could seed a new line of work at the intersection of vision, generative models, and intuitive physics.

*Methodological quality.*  The authors deliver a surprisingly effective, training-free baseline.  By measuring how *diverse* a diffusion model’s inpaintings become when each object is masked out, they extract a proxy for structural importance that beats size, depth, convexity, and other simple cues.  Experiments span COCO, NYU-v2, and a purpose-built “ClutteredParse” set, and the evaluation protocol (pairwise dependency accuracy plus full deconstruction score) is transparent and replicable.

*Clarity.*  The paper is easy to follow.  Diagrams show the step-by-step pipeline (mask → inpaint → CLIP/DINO diversity → rank → remove) and qualitative sequences make successes and failures intuitive.


**Weaknesses**

*Physical realism is only skin-deep.*  “Plausibility” is judged by a diffusion model’s ability to hallucinate sensible pixels, not by any physics engine or human study.  The system can therefore declare victory on scenes that *look* stable yet would collapse in reality (e.g., a table leg removed but magically inpainted).

*Dataset scope.*  The benchmark is built from repurposed still images.  Real-world support relations—complex stacks, articulated furniture, liquids—are rare.  Without video or 3-D data, the task avoids many of the hardest physical ambiguities, so it is unclear how far results generalize.

*Computational load and sensitivity.*  Calling a large diffusion model and dual feature encoders for *every* object is expensive.  There is no ablation across diffusion backbones or clip variants; performance may hinge on specific pretrained models or prompt engineering.

*Missing stronger baselines.*  No learning-based alternatives are tried—e.g., graph neural networks trained on synthetic physics, or models fine-tuned to predict support graphs.  Without them, we cannot tell whether the zero-shot method is truly hard to beat or merely faces weak competition.

*Temporal blindness.*  The task assumes a single static frame.  Some dependencies—stack order, friction, recent motion—cannot be inferred from a snapshot, capping the ceiling for any image-only approach.  A discussion of this limit and possible video extensions is needed.

Taken together, the paper is creative and thought-provoking but would benefit from tighter physical validation, a broader dataset, and stronger comparative baselines to establish lasting impact.

---

> ### Author Rebuttal · Authors · 2025-07-31
>
> We thank you for your insightful feedback and highly positive assessment. We are thrilled you found our work "**creative and thought-provoking**" and that the Visual Jenga task is "**genuinely novel**" with the potential to "**seed a new line of work**."  In general, we are thrilled that the reviewers unanimously found the task to be novel and significant.
>
> We now address the reviewer's concerns and questions.
>
> **“Physical realism is only skin-deep.”** and **“Plausibility is judged by a diffusion model’s ability to hallucinate sensible pixels, not by any physics engine or human study.”**
>
> Visual Jenga is a **single-image** scene understanding task, in the tradition of single view image understanding in computer vision (e.g. \[30,31,33,70\]) and psychology (e.g. \[9\]).  Recovering explicit 3D geometry and physical properties (e.g., mass, friction) from a single image is an ill-posed problem. Our goal is to investigate the limits of what can be inferred from a single static image without the benefits of information such as scene geometry and physical properties, or motion, similar to how humans understand still photographs. Just like with humans, this understanding will never be perfect.
>
> In this paper, we propose a simple, correlation-based baseline to tackle the Visual Jenga problem. Yes, sometimes the inpaintings by the model may be incorrect due to the current model’s limitations.  Despite this, the proposed method performs surprisingly well, as demonstrated by the human evaluation on full-scene decompositions.  Of course, we hope that the subsequent works will explore other, more sophisticated methods, including physics engine simulation. But to the best of our knowledge, current such methods \[46,71\] are not yet able to work on real-world images, as they are often designed for simplified or synthetic environments. We hope that the Visual Jenga task will encourage more work in this direction.
>
> **“Temporal blindness. The task assumes a single static frame. Some dependencies—stack order, friction, recent motion—cannot be inferred from a snapshot”**
>
> As discussed above, in this paper, we are exploring the single-image scene understanding setting.  However, a “Video Jenga” task—where a dynamic scene is observed before object removal—sounds like a great future work direction. Thank you for the suggestion.
>
> **“Have you verified that high-score removals are truly stable (e.g., human ratings or physics-engine tests)?”**
>
> Yes, we used human evaluators to assess physical plausibility in our full-scene decomposition evaluations (Section 4.2). If high-score removals resulted in implausible or unstable scenes, they would have been flagged as failures by the human raters.
>
> **“Dataset scope. The benchmark is built from repurposed still images. Real-world support relations—complex stacks, articulated furniture, liquids—are rare.”** and **“Any plan to add images or videos with denser, load-bearing stacks?”**
>
> We agree that, like in other computer vision tasks, increasing dataset complexity will be important moving forward. We believe it will become a community-driven effort, similar to recognition and detection tasks. However, we want to point out that our submitted datasets already contain a number of examples of load-bearing stacks (20% of the images in a full scene decomposition dataset have stacks of at least 4 layers). For example, Figure 7 features stacks of tires and a complex arrangement of books and glasses. The animated results in our supplementary video further demonstrate the method successfully deconstructing stacks of boxes (0:07), a ladder leaning on books (0:46), and other complex support structures.
>
> **“What is the per-image compute time?”**
>
> Approximately 3 minutes per object (on a single A6000 NVIDIA GPU) for the whole pipeline, including segmenting, inpainting, and removing. The time per-image depends on how many objects there are in the image.
>
> **“There is no ablation across diffusion backbones or clip variants; performance may hinge on specific pretrained models or prompt engineering.”**
>
> \- **Feature Encoder Ablation**: In Table 3 in the main paper and Section E (Figs. 13-14) of the supplement we show ablations between using only CLIP vs. only DINO vs. both, showing that the combination works best.  We have not ablated CLIP vs. OpenCLIP, etc.
>
> \- **Diffusion Backbone Ablation:** Diffusion Backbone Ablation: This is an excellent question. Our choice of the Runway Stable Diffusion 1.5 inpainting model was deliberate. A core principle of our approach is to be text-free, relying only on visual context. Most modern inpainting models are heavily text-dependent and fail to produce contextually meaningful results without specific text prompts. From our experiments, only the Runway’s model generated contextually meaningful results in a text-free manner, which our approach requires. For instance, other inpainting models we tested often generated completely irrelevant objects (like human faces) where a tabletop object was removed. We will include this discussion in the paper.
>
> **“Missing stronger baselines. No learning-based alternatives are tried—e.g., graph neural networks trained on synthetic physics, or models fine-tuned to predict support graphs.”** and **“Without them, we cannot tell whether the zero-shot method is truly hard to beat or merely faces weak competition.”**
>
> We are not aware of any existing learning-based baseline methods that would be appropriate for the Visual Jenga task and could be easily run – but we are open to suggestions.  However, we do provide the go-to learning baseline of 2025 – ChatGPT 4o in our supplementary.   We tested three distinct pipelines using ChatGPT 4o (with 4o image generation, InstructPix2Pix, and Molmo+SAM+Firefly, similar to ours) and found that ChatGPT struggles with the spatial grounding required for this visual task, especially in cluttered scenes. In contrast, our vision-first, counterfactual inpainting method bypasses this fragile text-to-vision grounding problem. We believe these comparisons reasonably address the concern. We will add a summary to the main paper for clarity.
>
> **“The authors should discuss safeguards and ethical use guidelines.”**
>
> We will add a paragraph to the discussion acknowledging that methods for plausible object removal could be misused to create fake image modifications that could be used for misleading visuals.

---

### Decision · Program_Chairs · 2025-09-17

**Decision:**

Accept (poster)

**Comment:**

The paper introduces a new task to judge scene understanding -- visual jenga that progressively removes the objects from a scene. This is accomplished by leveraging off the shelf models.

Initially, the reviewers agreed on the interestingness and value of the introduced task, found it original and simple. The reviewers had questions around learning-based comparisons, use of a simplified pipeline (VLM only) and were seeking clarifications around the correlation and causation.

The authors have addressed the initial concerns of the reviewers, and during the discussion all the reviewers recommended the acceptance. AC agrees with the reviewers and recommends to accept.